# Explaining refugee flows. Understanding the 2015 European refugee crisis through a real options lens

Linda Peters[1], Peter-Jan Engelen[2,3], Danny Cassimon [4]*

1 Statistics Netherlands (CBS), The Hague, The Netherlands, 2 Faculty of Business and Economics, University of Antwerp, Antwerp, Belgium, 3 School of Economics, Utrecht University, Utrecht, The Netherlands, 4 Institute of Development Policy (IOB), University of Antwerp, Antwerp, Belgium

* danny.cassimon@uantwerp.be

## Abstract

In 2015 the unprecedented arrival of refugees in Europe posed serious challenges for the EU and its member countries on how to deal with such an influx. A key element in better managing refugee flows is to understand what drives these flows in a certain direction. A refugee who comes to Europe has to make trade-offs in terms of cost and benefits, duration, uncertainty and the multi-staged character of the journey. Real options models are a suitable tool for modelling these kind of decision dynamics. On the basis of a case-study, that compares three routes from Syria to Europe, we demonstrate how well the real options analysis is in line with the development of the refugee flows.

## 1. Introduction

In 2015 the European refugee crisis began when the flow of migrants increased dramatically from 153,000 in 2008 to more than 1 million in 2015 [1]. This was mainly due to the growing number of Syrians, Iraqis, Libyans, Afghans and Eritreans fleeing war, ethnic conflict or economic hardship. With the exceptional volumes of new arrivals, an adequate response from Europe as one union was required, because the magnitude of the crisis was too large to solve for individual member states. For example, frontline states such as Greece and Italy bore a disproportionate responsibility for receiving new arrivals, transit countries such as Hungary and Croatia suddenly faced enormous pressure at their borders and the wealthier EU countries such as Germany and Sweden cope with the huge influx of refugees, because these are favored final destinations for migrants.

A major question that arises here is how the EU, individual countries and other stakeholders should address and manage this problem. A starting point for answering this question is to be able to better predict refugee flows. An assessment framework that allows to better understand the ways in which individuals process information, think through their options, and select courses of action is a key prediction tool. This is important, because we would like to know why migration movements progress the way they flow and try to understand why certain routes are more preferred by refugees to take than other routes.

**Competing interests:** The authors have declared that no competing interests exist.

The flows themselves are extremely complex and driven by a broad range of conditions in the countries of origin, transit and destination, and in the relationships between them [2]. The flows consist of a variety of individuals and families such as asylum seekers, war refugees, climate refugees, stateless persons, labor migrants and economic migrants, who come to Europe through authorized as well as irregular channels for various reasons. The identification and the legal differences and consequences are outside the confines of this article. In this article, we use refugees and migrants as a generic term for all groups. The basic idea here is that refugees or migrants are coming to Europe and that these individuals have to make trade-offs in terms of costs and benefits, uncertainty, the duration and the the multi-stage character of the journey in order to make a decision on the best route to travel to Europe. A potential refugee has to make a careful deliberation whether or not to make the crossing, at which time it happens, in what way, according to which route, etc. Nowadays, migrants keep themselves informed of the developments that could have an impact on their journey and adjust their decisions on a real-time basis [1]. Especially, when the refugee journey takes place in several phases, and the refugee does not arrive straight away on the desired final destination, they have several decision options at their disposal. In that case they make the actual decision at the moment they arrive at a transit point, based on the particular situation of that specific moment.

These dynamics could be captured in a real options framework, which is illustrated on the basis of a case-study. This case-study will investigate whether the predictions of the real options framework match with the refugee volume data. We assess three important routes, the Central Mediterranean Route, the Eastern Mediterranean Route by land and the Eastern Mediterranean Route by sea, to Europe in the years 2014 and 2015 and verify whether the model's predictions follow the same pattern as the volumes. Our starting point of the case study is the situation of a Syrian migrant that already has taken the decision to leave Syria. The refugee needs guidance for the decision between the different routes available through several countries to the desired final destination and which routes it is best to take. The main objective of this paper is a first attempt to model the dynamics of the decision-making process of the average refugee using the real options framework. We try to understand why certain routes are more interesting for the migrant to take than other routes.

The article is structure as follows. In section 2, we discuss refugee routes as a real option and section 3 presents the value drivers of those routes. Section 4 provides an application of real options to the 2015 refugee crisis. This case-study provides a comparison of three routes from Syria to Europe that investigates how well the real option logic is in line with the evolution of the observed refugee flows at the time. Finally, section 5 contains the conclusions.

## 2. Modeling refugee flows

Refugees coming to Europe have to make trade-offs in terms of cost and benefits, duration, uncertainty and the multi-staged character of the journey. The choice for the best route between alternative paths by the refugee is analogous to make a selection between several potential investments. The decision of the refugee indeed shows similarities to that of an investment decision under uncertainty. Hence, a real options framework is a suitable tool for modelling these kind of dynamics.

Early work along these lines focuses on labor migration. Sjaastad [3] is the first to acknowledge that it could be viewed as an investment. Todaro [4] focuses on the wage differential between the host country and the country of origin as the main variable affecting labor migration. Burda [5, 6] models migration as an investment decision under uncertainty, which is built on the ideas of Dixit and Pindyck [7]. His work refers to the fact that a migration decision will also depend on the value of waiting and found that when a migration decision is

postponed, it generates a positive value if there is uncertainty about the future wage differences. Locher [8] explores a similar concept in a two-period framework, using data on ethnic German migration from CIS countries (Russian Commonwealth). Khwaja [9] has extended the framework of Burda [5, 6] by describing the role of uncertainty in the migration decision, while Bayer and Juessen [10] model internal migration decisions in the United States.

The above-mentioned literature models a migration decision as a simple call option, where the labor migrant has the possibility, but does not have the obligation to migrate. The migrant has the option to wait if relevant information can be expected to reveal itself over time to take a better informed decision. For example, it could be profitable to postpone the migration decision, because the migrant is expecting 'bad news', or because the sunk cost could be decreasing. Most of the existing literature models migration decisions as a simple call option for straight one-off moves. One exception where labor migration is modeled as a compound option is provided by Artuc and Ozden [11]. The authors construct a multi-period model of dynamic transitory migration decisions, where the utility of living in a particular destination is linked to the option value to migrate further. In their model the value of the option to wait is derived from the underlying volatility of the economic environment. Even though the dynamic discrete choice model of migration is a multi-period model, it assumes that a migrant travels through legal channels, where the decision is voluntarily and based on the option to wait.

However, current models do not meet the requirements to model the decision-making process of a refugee during the 2015 European crisis. A journey of such a refugee contains a mix of involuntary and voluntary decisions, involves multiple stages, through several countries, and proceeds either through regular or irregular channels or both, where at each stage the refugee has to make a trade-off between the costs and benefits, uncertainty and duration, in order to choose the best route for reaching the desired destination. A multiphase real options framework is therefore needed in order to model the multistage character of refugee routes, where each leg of the specific route is a phase of the model. Therefore, in this article we model the dynamics of the multiphase decision-making process of the average refugee through the use of a sequential real option model. In contrast to the aforementioned models, where the labor migrant could choose to postpone the decision to migrate, we assume that the refugee has already decided to flee, but has to choose between different routes through which to reach its final destination.

We model the choice of a refugee for certain routes as the choice between multiple costly decisions. Most refugees are able to make a reasonably rational choice based on the information available to them on characteristics of the various routes. We expect refugees to consider information on particular factors, such as the desired final destination, the expected costs and benefits, the time and the risk of reaching their final migration destination. Even in case of war refugees, where the first stages of the journey often have a forced character and where consecutive stages are characterized by a more voluntary character, refugees can still actively make choices regarding routes [12]. For example, on a number of Facebook pages, refugees are able to find precise information on smuggler services revealing concrete prices and departure points. In addition, the Facebook pages also include the facilitation on organizing the logistics needed for the travel [1].

The objective of our article is to 'value' migration routes, which is analogous to the valuation of an investment decision under uncertainty with irreversible investment costs, and takes into account the various factors that have an impact on the decision of the migrant. To value the attractiveness of a certain route we use the multi-phase compound options framework by Cassimon et al. [13].

Compound options have been widely used in the financial literature to evaluate sequential investment opportunities. Such an investment process consists of a sequence of investment

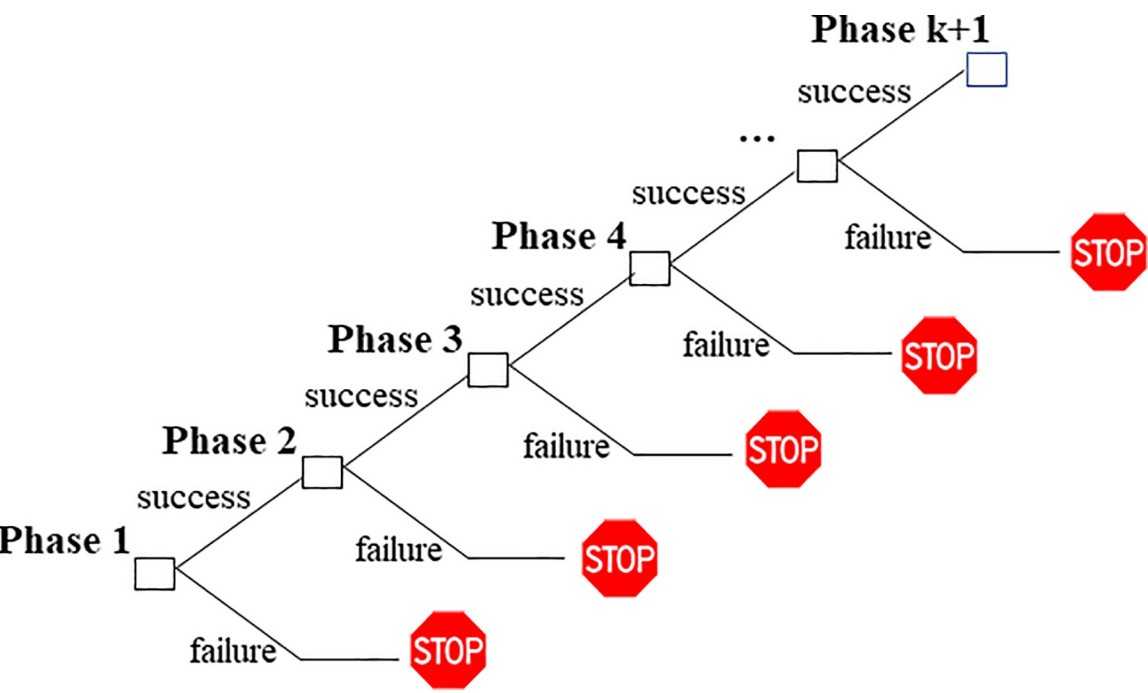

**Fig 1. The optional nature of a sequential investment project.** Source: Authors' own elaboration.

phases, in which each phase creates an option for moving to the next phase. If a previous-stage turns out to be successful, the next one will be initialized; otherwise the investment process is discontinued. This process goes on until the final stage (Fig 1). Let us consider an investor who wants to invest in a project whose commercial phase cannot be launched upon the successful competition of previous $k$ investment stages. Let $T_{k+1}$ be the time of the market launch, when, upon paying the commercialization cost $I_{k+1}$, the firm earns the project value $V$. The project payoff at time $T_{k+1}$ is $\max[V-I_{k+1}, 0]$. Let $C_{k+1}(V, t)$ denote the value at time $t$ of this 1-fold compound option or single stage investment opportunity. We assume that the commercialization phase is reached upon investing an amount $I_k$, at time period $T_k$, with $T_{k+1} \geq T_k \geq \cdots T_2 \geq T_1 \geq 0$. The project starts with $I_1$ as the startup costs, while $T_k$ and $I_k$ are maturities of intermediate phases that lead up to the commercialization phase and the respective investment costs. At any stage $k$ the investor can decide to abandon the project or to enter the next stage, hence, the optional nature of the investment project. Each stage therefore creates an option on the next stage. In this way the investment problem becomes a chain of options. Compound options typically capture the value of such a multi-stage investment project well.

Cassimon et al. [13] develop a generalized $N$-fold compound option model that explicitly incorporates both commercial (market) and technical uncertainty to value sequential multi-stage investment projects. Technical uncertainty refers to technical success of each investment stage by multiplying the options value at each decision point with the probability of technical success at that stage. In this model, the project has a commercial risk $\sigma$ and technical success probabilities $p_1, p_2, \ldots, p_{k+1}$ at each investment stage. The project value is unknown and is denoted by $V_t$ at time $t$. It is described by a Geometric Brownian motion $dV_t = \mu V_t dt + \sigma V_t dW_t$, where $\mu$ and $\sigma$ represent the growth rate and the standard deviation of the project value.

If one defines a sequence of call options with value $C_k$ on the call option whose value is $C_{k+1}$ with exercise price $I_k$ and expiry date $T_k$, such that

$$C_k(C_{k+1}(V, T_k), T_k) = p_k max[C_{k+1}(V, T_k) - I_k, 0], \tag{1}$$

where $C_{k+1}(V, T_k)$ stands for the value of the underlying compound option. The pricing formula for the $k+1$-fold compound option, $C_1(V, 0)$ at $t = 0$, is given by the following expression:

$$C_1(V, 0) = h_{k+1} V N_N(a_1, a_2, \ldots, a_N; \ R_1^{k+1}) - \sum_{l=2}^{k+1} h_l! I_l e^{-rT_l} N_l(b_1, b_2, \ldots, b_l; \ R_1^l)$$
$$+ -h_1 I_1 e^{-rT_1} N_1(b_1), \tag{2}$$

where

$$a_l = b_l + \sigma \sqrt{T_l}; \ \ l = 2, \ldots, k+1 \tag{3}$$

and

$$b_l = \frac{ln\frac{V}{V_l^*} + (r - \frac{\sigma^2}{2})T_l}{\sigma \sqrt{T_l}}; \ \ l = 2, \ldots, k+1 \tag{4}$$

$V_l^*$ (i.e., the threshold level for immediate start or continuation of the next phase) is the at-the-money option solution of

$$C_{l+1}(V, t_l) = I_l; \ \ l = 1, \ldots, k+1 \tag{5}$$

$$\rho_{fg} = \sqrt{\frac{T_f}{T_g}}; \ \ 1 < f < g \le k+1 \tag{6}$$

$$R_1^l = (a_{fg}^l)_{f.g=1,2,\ldots,l} \text{ with } \begin{cases} a_{ff} = 1 \\ a_{fg} = a_{gf} = \rho_{fg} \end{cases}; \ \ 1 < f < g \le k+1 \tag{7}$$

and

$$h_{k+1} = p_1 p_2 \ \cdots \ p_k p_{k+1} \tag{8}$$

$$h_k = p_1 p_2 \ \cdots \ p_k \tag{9}$$

$$h_2 = p_1 p_2 \tag{10}$$

$$h_1 = p_1. \tag{11}$$

This compound option model will be used to model the decision of a refugee (e.g. from Syria) who wants to move to a final destination country (e.g. Germany). In order to do so, this individual needs to pass through several interim countries (e.g. Libya, Italy). Successfully reaching the final destination depends on the successful completion of previous $k$ stages. Each stage (e.g. moving from Syria to Libya) will come with certain costs and benefits, will require a certain amount of time and will have a certain probability that the refugee gets stuck in the interim country. Refugees will compare several routes based on those characteristics. The route with the highest option value, would be the best choice, and would be the most appealing route for the refugee. Note that we are not so much interested in the absolute value itself, as the absolute real option value of a particular route does not mean a lot by itself. Rather, we are

particularly interested in the relative value of the potential routes as this will indicate whether a certain route is more attractive than another. Hence, it will allow for a better understanding of routing choices.

As such, each of the parameters of the compound option model determines the value of a migration route. The expected benefits at desired destination $V$ are the benefits which the migrant expects to obtain when he or she reaches the desired final destination. These benefits in the destination country might include the right to work and live in the host country and, therefore, to receive access to education, private housing, healthcare, employment opportunities and social assistance, in addition to the basic support. Alongside the benefits and costs at the desired destination, there are also benefits and costs for the intermediate locations. In the neighboring and transit countries the benefits serve the basic needs such as food and shelter. The costs include the costs of traveling to the next destination point. The next parameters of an intermediate phase are $T_i$, the time horizon in days, which refers to the time that a refugee needs to move to the next phase and the probability $p_i$ the refugee will be able to make it to the next phase. For instance, the placing of a border fence has an impact on the probability of not arriving at the next phase. The parameter $\sigma$ represents the uncertainty of benefits at the desired destination, which for example depends on the changing asylum policy in a given country.

## 3. Case study of the 2015 European refugee crisis

This section will illustrate our model by focusing on a case study, since there is no data on a large sample available. This case-study demonstrates the modelling of refugee decision-making through a real options framework. The objective of this case study is to illustrate that the popularity of the different routes runs parallel with their real options values, i.e. to show that a route becomes more popular as soon as the real options value of that route increases relative to other routes, and that a route becomes less popular when the real options value decreases relative to other routes. This is performed on the basis of the situation of the average Syrian adult male migrant who fled to Europe and had to choose between three main routes: Central Mediterranean Route (CMR), Eastern Mediterranean Route by sea (EMR by sea) and Eastern Mediterranean Route by land (EMR by land). In 2013 and 2014, the CMR was slightly favored. However, during the European refugee crisis of 2015, Syrians avoided the CMR and the EMR by sea became by far the most popular one, dwarfing the migrant volumes of the other routes. We will demonstrate that this turning point is clearly visible in the real options value of the routes.

### 3.1. Description of the main routes from Syria to Europe

As a consequence of the Arabic Spring in December 2010 in countries such as Tunisia, Egypt and Libya, a revolt started in Syria in March 2011. In September 2016, about 11 million Syrians had been displaced from their homes since the start of the Syrian war. More than half of them, about 6.6 million, were internally displaced within Syria. Furthermore, 4.8 million refugees have fled to Turkey, Lebanon, Jordan, Egypt and Iraq. In addition, about 1 million Syrians crossed into Europe to seek asylum. The most popular destinations within the EU were Germany with 300,000 asylum applications and Sweden with 100,000 asylum applications.

Syrian refugees have mainly used three routes for their journey to Europe: Eastern Mediterranean route by land (EMR by land), Eastern Mediterranean route by sea (EMR by sea) and Central Mediterranean route (CMR). The three routes are presented in Fig 2.

The refugees using the EMR by sea arrived on several Greek islands, most on Lesbos, while others have entered Greece via the land border, or else exited Turkey directly into southern Bulgaria (EMR by land). Most of the refugees continued their journeys north, leaving Greece and Bulgaria through its border with the former Yugoslav Republic of Macedonia via Serbia

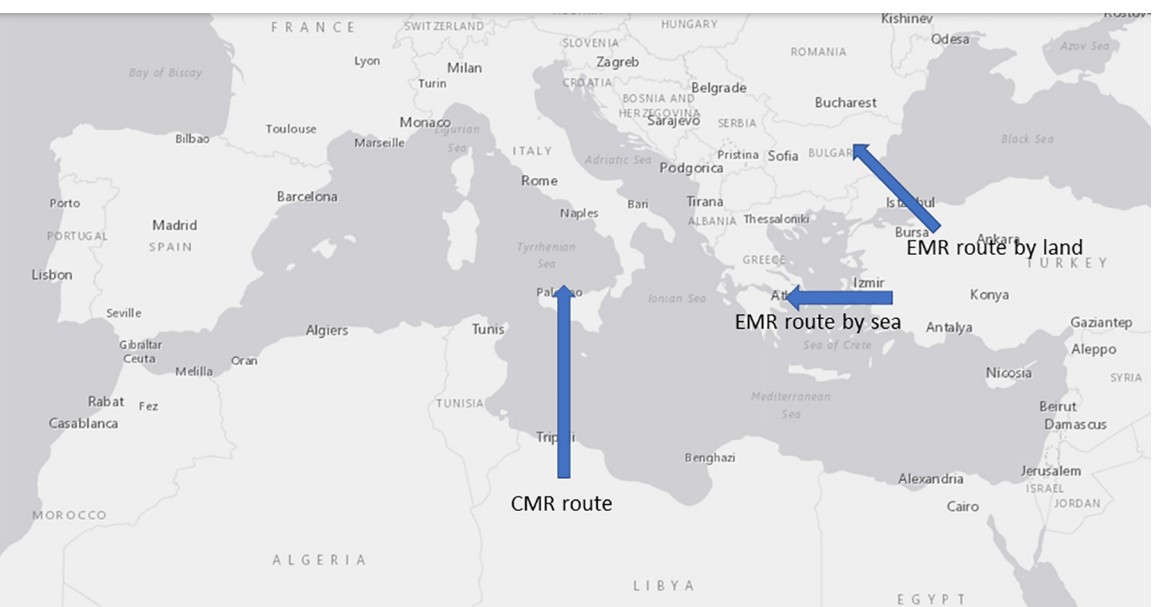

**Fig 2. Three main routes into Europe.** Source: Authors' own elaboration based on USGS National Map Viewer.

into Hungary and Croatia and then towards Western Europe. This is the so-called Western Balkan route. The CMR is the route over sea from North Africa, mostly from Libya, to Italy and Malta. Although the crossing of the CMR was long, Italy remained a preferred destination over Greece until 2014 because migrants arriving in Greece had to pass through the Balkans to get to Germany. However, this changed in 2015. Syrians started to avoid Libya due to the deteriorating political and security situation in Libya, increasingly poor conditions in transit countries and the perception that the EMR was relatively safer [14]. Furthermore, Egypt had blocked the border to Libya from mid-2014 on, and Algeria removed visa free travel arrangements at the end of 2014 [15]. Due to the construction of border fences by Greece in 2012 and Bulgaria in 2014 along their borders with Turkey [16], the EMR by sea was more popular than the route by land. This is shown in Fig 3. For example, one can see that in 2014, 50% of the Syrians took the CMR. However, in 2015 this changes drastically, only 1% is taking the CMR, whereas 82% are taking the EMR by sea to Greece.

## 3.2. The real option value of each route

Each route is modeled as a multistage compound option as the (Syrian) refugee has to travel through several countries (geographical legs) before arriving in Germany (final destination). Fig 4 represents a refugee route as a (k+1)-stage compound option. Each geographical leg describes the phase between two geographical locations, for example between the starting point and the first transit point, or between the last transit point and the final destination. A geographical leg begins at the moment a refugee departs from the start location of that leg and ends with the stay at the final location of that leg. In Germany the refugee has to apply for asylum and therefore ends up in the asylum procedure; here, we refer to both "asylum"according to the German constitution as well as „refugee status"according to the 1951 Convention. The asylum procedure is modelled as an additional fourth phase as it is uncertain whether or not the refugee can stay in Germany. Finally, once the asylum has been granted, the refugee is able to build a life in Germany during this final phase. For this final phase, we need the expected benefits and its uncertainty. In order to estimate the value for these value drivers, we make use

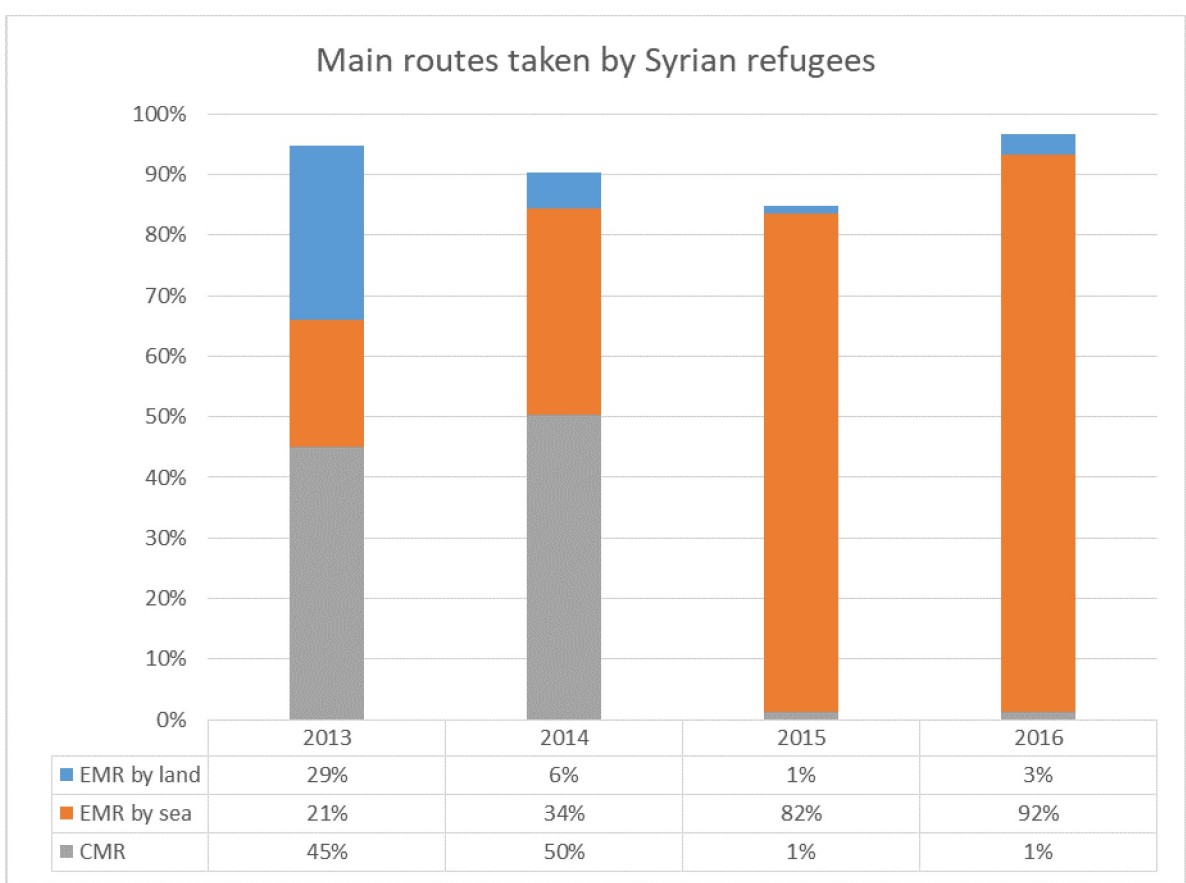

**Fig 3. Illegal border crossings detected by Frontex (Syrians only) of three main routes.** Central Mediterranean Route (grey), Eastern Mediterranean Route by sea (orange) and Eastern Mediterranean Route by land (blue). Source: [17].

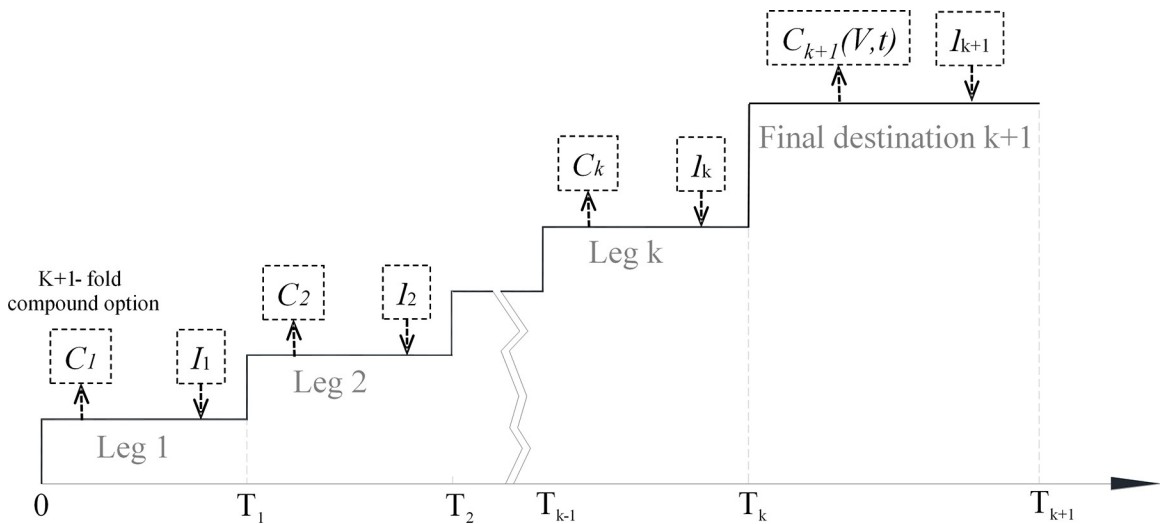

**Fig 4. A refugee route as a k+1-stage compound option.** Source: Author's own elaboration.

of publications from *Destatis*, the Federal Statistical Office of Germany [18]. For the asylum procedure phase, we use the data from the *Asylum Information Database* (AIDA) [19]. For the geographical legs we take as a proxy for the value of the benefits of the leg the available monthly financial allowance/vouchers granted to asylum seekers at the final location of a leg from the country reports on the AIDA. It is not always clear whether or not refugees actually do receive these benefits (in cash or kind), but we believe this is a reasonable proxy of the possible benefits of a particular leg.

In order to estimate the probability of arrival, we use volume figures of AIDA [19], *International Organization for Migration* (IOM), UNHCR [20] and Frontex [17]. Because of the often clandestine character of refugees, there are very few comprehensive sources available for the other value drivers. Therefore, we rely on information available through press, Facebook and other social media. An important aspect to consider here is the length of a leg: the time needed for a refugee in order to bridge the leg. In the public anecdotal information, often the net travel time is denoted. In case we would use this travel time for the length of a leg, we assume that the refugee is able to travel without any obstacles or delays whereas the refugee has to find the way to a new place or has to find the route to the next transit stop. Since this is not very likely to be the case, we correct for this and use for the length of a leg twice the travel time that is reported in the public anecdotal information. In this way, we are not only trying to provide realistic durations for each specific leg, but also consider the duration of the journey as a whole (see for example [21]).

**3.2.1. Route 1 –The Central Mediterranean Route.** In Table 1 the values of the parameters for the Central Mediterranean Route in 2014 and 2015 are presented, i.e. before and after Libya became less popular as a transit country, for the Syrian refugee who flees to Germany; for converting the values from dollar to euro we have used the OECD yearly average exchange rates [22]. Firstly, the route consists of three geographic legs: from Syria to Libya, from Libya to Italy and from Italy to Germany. Upon arrival in Germany, the refugee enters the asylum procedure by applying for asylum. Finally, the final phase is the temporary (3-year) stay of the refugee in the desired destination country Germany.

In 2014 the journey from Syria to Libya was mostly travelled by road. The route passes through Jordan, across the Sinai desert and ends in the Egyptian border town of El Salloum [23]. A refugee payd almost 400 euro per person to the smuggler for this 5-day journey. This is the irreversible cost or exercise price in our model to reach the next phase. (See variable $K_1$ in Table 1). In Libya, Syrians were in general well-received and regarded as fellow-Arabs. Most of the times they searched for a job or tried to travel directly to the next destination. Since Syrians might have to wait a few days up to several weeks [24] to search for a smuggler who could assist them with the crossing to Italy, we set the length of travel at 24 days for both 2014 and 2015 (see variable $T_1$ in Table 1). We put the benefits that Syrians receive during their stay in Libya to zero. The powerlessness of the Libyan government to secure its borders and the presence of a vast desert territory made it easy for Syrians to enter the country through the use of smugglers. As a reasonable approximation for this, we set the probability of arrival to 85% in 2014 (see variable $p_1$ in Table 1). At the end of 2014, the situation in Libya deteriorated and Algeria and Egypt closed their borders with Libya. Syrians were still able to enter Libya by catching a plane to Algeria and consecutively cross the border by using a smuggler at a price of 300 euro [25]. However, there are not many Syrians who did this. The UNHCR Statistical Database indicates that in 2014 18,653 Syrian refugees resided in Libya, whereas in 2015 there were only 97 refugees [26]. Therefore, we set the probability of arrival for 2015 at a very low value, 2%.

The journey continued by sea to Italy. In 2014 smugglers asked 2,000 euro for this journey [1] and in 2015 they asked between 800 and 3,600 euro [27]. According to the IOM, there were

**Table 1. Value drivers and real options value Central Mediterranean Route in 2014 and 2015.**

| Stage | Parameter | Symbol | Unit | Value 2014 | Value 2015 |
|---|---|---|---|---|---|
| Leg from Syria to Libya | | | | | |
| | Benefits of travel | v_1 | euro | 0.00 | 0.00 |
| | Costs of travel | K_1 | euro | 377.00 | 400.00 |
| | Length of travel | T_1 | days | 24 | 24 |
| | Probability of arrival | p_1 | | 85% | 2% |
| Leg from Libya to Italy | | | | | |
| | Benefits of travel | v_2 | euro | 0.00 | 0.00 |
| | Costs of travel | K_2 | euro | 2000.00 | 2500.00 |
| | Length of travel | T_2 | days | 18 | 18 |
| | Probability of arrival | p_2 | | 75% | 75% |
| Leg from Italy to Germany | | | | | |
| | Benefits of travel | v_3 | euro | 77.47 | 77.47 |
| | Costs of travel | K_3 | euro | 500.00 | 750.00 |
| | Length of travel | T_3 | days | 7 | 7 |
| | Probability of arrival | p_3 | | 70% | 70% |
| Asylum procedure in Germany | | | | | |
| | Benefits | v_4 | euro | 600.60 | 457.60 |
| | Costs | K_4 | euro | 580 | 442 |
| | Length | T_4 | days | 126 | 96 |
| | Refugee rate | p_4 | | 85.9% | 99.5% |
| Germany | | | | | |
| | Annual benefits | V | euro | 14041 | 14041 |
| | Annual costs | I | euro | 3239 | 3239 |
| | Volatility of benefits | sigma | | 22% | 25% |
| | **Net real options value** | ROV | euro | 10388 | -113 |

Source: prepared by the authors

2,892 fatalities in 2015 on the Central Mediterranean Route [28] and 153,842 arrivals in Italy [29]. The survival rate of the crossing will be 153842 / (153842+2892) = 98%. However, this is not the same as the probability of arrival, since for example it is not taken into account how many persons have been sent back or were not able to obtain a spot on a boat. Therefore, we set the probability of arrival in Italy at 75%. Since there is no different data on 2014, we set this in the base case equal to 2015. As a proxy for the benefits that the refugees receive in Italy, we need to take the financial allowances/vouchers to asylum seekers in Italy. These benefits are not present, therefore we set their benefits to 0 euro [30, 31]. The boat journey usually takes two to six days [24]. By law, asylum-seekers can be held for up to a maximum of a month in an accommodation center for asylum seekers [32]. Therefore, as an approximation we set the duration to 18 days for crossing the Mediterranean Sea and the waiting time in Italy.

The final 'geographic' leg is the leg to Germany. In 2014, for example, an amount of 4,000 euro was paid in order to travel with a transporter for eight people to Germany, which comes down to 500 euro per person, and in 2015 amounts between 500 and 1,000 euro per person were generally accepted [33]. Therefore, we set it equal to 750 euro. The duration for this journey by car, including the search of a smuggler, is estimated to take seven days. As a proxy for the benefits, we take the financial allowances to asylum seekers in Austria, which comes down to 332 euro per month [34], in other words 332 x 7 / 30 = 77.47 euro for four days. It is not always clear whether or not refugees actually did receive these benefits (in cash or kind), but

we believe this is a reasonable estimation of the possible benefits of a leg. Finally, we set the probability of arrival to 70%.

During the asylum procedure in Germany, the allowance in the reception center including food amounted to 143 euro per month [35]. The average duration of the asylum procedure for Syrian asylum seekers in Germany was 4.2 months in 2014 and 3.2 months in 2015 [36], therefore the benefits for Syrian refugees to travel to Germany equal 4.2 x 143 = 600.60 euro in 2014 and 3.2 x 143 = 457.60 euro in 2015. We have taken the *Existenzminimum* (single person), the minimum payment for survival such as for accommodation and heating, as an indication for the costs that asylum seekers have to pay by themselves. However, the amount that an asylum seeker needed is lower in comparison to the *Existenzminimum* due to the following two reasons: Firstly, we assume that the asylum seeker stays in an asylum seekers center, which results in lower costs. Secondly, the asylum seeker received several benefits, such as food and shelter. We correct the minimum level of subsistence for both factors and take the figures from 2015 as a proxy for 2014 and 2015.

In 2015, the *Existenzminimum* was 8,472 euro per year [37], or 8472/12 = 706 euro per month. The monthly allowance for a single adult was 143 euro for staying in an accommodation center and 359 euro for staying outside an accommodation center [38]. We apply this ratio in order to correct the minimum level of subsistence for staying in an accommodation center: 706 x (143/359) = 281 euro. Since the refugee received 143 euro per month for food, accommodation etc., the costs at the expense of the refugee were equal to 281–143 = 138 euro per month. For 2014, this will be equal to 138 x (126/30) = 580 euro for 4.2 months (= 126 days) and for 2015 this will be 138 x (96/30) = 442 euro. Finally, the probability of entitlement to asylum is based on country reports of Germany [38, 39]. From this we find that the refugee rate for Syrians, i.e. the percentage of applicants who receive the refugee status, was 85.9% in 2014 and 99.5% in 2015.

Once asylum has been granted to the refugee, Germany had to incorporate expenses for education, private housing, health and employment opportunities and social assistance [18, 40]). The individual federal states bore the majority of these costs [40] and we take these costs [18] as a proxy for the benefits that the refugee receives in Germany. We also assume that the differences in the costs per person within a federal state is limited and that the volatility of these costs for a major part is caused by the differences between the federal states. Based on this we derive that in 2014 the annual expected benefits for a refugee in Germany were equal to an annual amount of 6,570 euro with a volatility of 22% [18]. When we perform this calculation for 2015, we arrive at annual benefits of 5,414 euro with a volatility of 25% [18]. However, there is a complicating factor at play here. The benefits for the refugee, which are equal to the costs of the federal states, appear to decrease in 2015 when compared to 2014, whereas in fact they did the opposite: they increased. A large part of the costs for the federal states in 2015 have been booked in 2016, which results in annual benefits of 30,139 euro in 2016 [18]. In order to provide a realistic picture, we take as a proxy for the annual benefits in 2014 and 2015, the average value over the years 2014, 2015 and 2016: (6570 + 5414 + 30139) / 3 = 14,041 euro. In addition to the benefits that the refugee received, he or she also has to incur certain expenses. These total necessary expenses are estimated on the basis of the minimum wage. The minimum wage in Germany was 1,440 euro per month in 2014 and 2016. From this we could note that the additional annual costs for a refugee were equal to 1440 x 12–14041 = 3,239 euro per year. In case a refugee is entitled to asylum, then he or she received a residence permit for three years. For this reason, we assume that the refugee could count on to receive expected benefits in Germany for three years.

We notice that mainly because there is a lower probability of arriving in Libya ($p\_1$) the real options value of the CMR decreased from 10,388 in 2014 to -113 in 2015. Besides this

lower probability also the irreversible cost of the different stages went up. For instance, the cost of stage 2 increased from 2000 to 2500 euro, while the cost of stage 3 went up from 500 to 750 euro. One can simulate the effects in isolation. If we increase the irreversible costs from 2014 to 2015, but keep the probabilities in 2015 at the previous year level, the real options value drops to 9,934. If we decrease the probability in 2015, but keep the irreversible costs at the 2014 levels, then the real options value drops to -80 euro. If both variables change simultaneously, the real option value drops to -113 euro.

**3.2.2. Route 2 –The Eastern Mediterranean Route by sea.** The values of the parameters for the EMR by sea in 2014 and 2015 are presented in Table 2. This route is also divided in three geographical legs: from Syria to Turkey, from Turkey to Greece and from Greece to Germany, and in addition there is the phase of the asylum procedure and the phase after the asylum has been granted to the refugee. In this case-study we only discuss the first three legs, as the values of the value drivers during and after the asylum procedure in the desired destination Germany are similar to those of the CMR. In Turkey, asylum seekers only received an allowance when needed [41, 42], for this reason we set the benefits to zero. In 2014, the amount that smugglers received from the refugees in order to travel to Turkey equaled 189 euro [43] and in 2015 it was 361 euro [44]. Since the refugees reportedly travelled though territory controlled by armed groups, we estimate the duration of the travel at 14 days. Since 2011 Turkey maintained a generous open door policy to Syrian refugees [45, 46]. Therefore, we set the probability of arrival at 85%.

**Table 2. Value drivers and real options value for Eastern Mediterranean Route by sea in 2014 and 2015.**

| Stage | Parameter | Symbol | Unit | Value 2014 | Value 2015 |
|---|---|---|---|---|---|
| Leg from Syria to Turkey | | | | | |
| | Benefits of travel | $v_1$ | euro | 0.00 | 0.00 |
| | Costs of travel | $K_1$ | euro | 189.00 | 361.00 |
| | Length of travel | $T_1$ | days | 14 | 14 |
| | Probability of arrival | $p_1$ | | 85% | 85% |
| Leg from Turkey to Greece | | | | | |
| | Benefits of travel | $v_2$ | euro | 0.00 | 0.00 |
| | Costs of travel | $K_2$ | euro | 1600.00 | 1353.00 |
| | Length of travel | $T_2$ | days | 14 | 14 |
| | Probability of arrival | $p_2$ | | 75.0% | 75.0% |
| Leg from Greece to Germany | | | | | |
| | Benefits of travel | $v_4$ | euro | 109.33 | 109.33 |
| | Costs of travel | $K_4$ | euro | 160.00 | 160.00 |
| | Length of travel | $T_4$ | days | 24 | 24 |
| | Probability of arrival | $p_4$ | | 25% | 25% |
| Asylum procedure in Germany | | | | | |
| | Benefits | $v_3$ | euro | 600.60 | 457.60 |
| | Costs | $K_3$ | euro | 580 | 442 |
| | Length | $T_3$ | days | 126 | 96 |
| | Refugee rate | $p_3$ | | 85.9% | 99.5% |
| Germany | | | | | |
| | Annual benefits | V | euro | 14041 | 14041 |
| | Annual costs | I | euro | 3239 | 3239 |
| | Volatility of benefits | sigma | | 22% | 25% |
| | **Net real options value** | ROV | euro | 3149 | 3809 |

Source: prepared by the authors

Consecutively the route continued by sea to Greece. In Greece refugees did not receive any financial allowances [47, 48]. For the sea travel itself, the refugees paid about 1,600 euro in 2014 [49] and about 1,350 euro in 2015 [27]. The journey itself just took a few hours, but since the refugees also needed to make sure to have a seat on the boat and this journey could fail a couple of times we estimate a duration of 14 days. In 2014 there were 72,632 arrivals in Greece [28] and 59 casualties [50] on this route, for which we estimate the survival rate on 72632 / (72632 + 59) = 99.9%. In 2015 there were 853,650 arrivals in Greece [29] and 806 casualties [28], which also results in a survival rate of 99.9%. Like for the CMR, we set the value at 75%. Finally, there is the leg from Greece to Germany. This route goes through Serbia and Hungary towards Germany. We estimated its duration to be 24 days, the costs of travelling to average 160 euro and the benefits to amount to 109 euro, while there is a probability of successfully moving to the next leg of 25%. With these values for the different parameters, the real options value of the EMR by sea equals 3,149 in 2014 and 3,809 in 2015.

**3.2.3. Route 3 –The Eastern Mediterranean Route by land.** Finally, Table 3 provides the values of the parameters for the EMR by land in 2014 and 2015. Again, this route is divided in three geographical legs: from Syria to Turkey, from Turkey to Bulgaria and from Bulgaria to Germany, and in addition, the phase of the asylum procedure and the phase after granting the asylum. We only discuss the leg from Turkey to Bulgaria of this route. As a proxy for the journey from Bulgaria to Germany, we take the journey from Greece to Germany, as was discussed

**Table 3. Value drivers and real options value for Eastern Mediterranean Route by land in 2014 and 2015.**

| Stage | Value driver | Symbol | Unit | Value 2014 | Value 2015 |
|---|---|---|---|---|---|
| Leg from Syria to Turkey | | | | | |
| | Benefits of travel | v_1 | euro | 0.00 | 0.00 |
| | Costs of travel | K_1 | euro | 189.00 | 361.00 |
| | Length of travel | T_1 | days | 14 | 14 |
| | Probability of arrival | p_1 | | 85% | 85% |
| Leg from Turkey to Bulgaria | | | | | |
| | Benefits of travel | v_2 | euro | 15.51 | 0.00 |
| | Costs of travel | K_2 | euro | 500.00 | 1000.00 |
| | Length of travel | T_2 | days | 14 | 14 |
| | Probability of arrival | p_2 | | 16% | 16% |
| Leg from Bulgaria to Germany | | | | | |
| | Benefits of travel | v_3 | euro | 109.33 | 109.33 |
| | Costs of travel | K_3 | euro | 160.00 | 160.00 |
| | Length of travel | T_3 | days | 22 | 22 |
| | Probability of arrival | p_3 | | 25% | 25% |
| Asylum procedure in Germany | | | | | |
| | Benefits | v_4 | euro | 600.60 | 457.60 |
| | Costs | K_4 | euro | 580 | 442 |
| | Length | T_4 | days | 126 | 96 |
| | Refugee rate | p_4 | | 85.9% | 99.5% |
| Germany | | | | | |
| | Annual benefits | V | euro | 14041 | 14041 |
| | Annual costs | I | euro | 3239 | 3239 |
| | Volatility of benefits | sigma | | 22% | 25% |
| | **Net real options value** | ROV | euro | 674 | 577 |

Source: prepared by the authors.

before. The only deviation is the duration of the journey, which we set at 22 days instead of 24 days. For the other legs we refer to the other routes earlier discussed in this case study.

The crossing from Turkey to Bulgaria by land was difficult, because in response to the refugee flows, Bulgaria had started to place fences along its border with Turkey since 2014. Therefore, we assume that the leg from Turkey to Bulgaria took 14 days. For this leg, refugees paid 500 euro [51] in 2014, whereas this has increased to 1,000 euro [52] in 2015. In 2014 about 38,500 people attempted to cross irregularly the Bulgaria-Turkey border, of which some 6,000 of them indeed have reached Bulgaria [53]. Therefore, we estimate the probability of arrival in Bulgaria to 6000 / 38500 = 16%. We also use this value for 2015. Once having arrived in Bulgaria, refugees received a financial allowance of 33.23 euro per month [54], however, this was reduced to zero in 2015 [55]. In 2014, we set the benefits of travel equal to 33.23 x 14/ 30 = 15.51 euro, whereas this is zero in 2015.

With these values for the value drivers the real options value of the EMR by land equals 674 in 2014 and 577 in 2015.

### 3.3. Discussion

The objective of this case study was to illustrate that the popularity of the different routes runs parallel with their real options value. A route becomes more popular as soon as the real options value of that route increases relative to other routes, and that a route becomes less popular when the real options value decreases relative to other routes. This was performed on the basis of the situation of the average Syrian adult male refugee who fled to Europe and had to choose between three main routes: Central Mediterranean Route (CMR), Eastern Mediterranean Route by sea (EMR by sea) and Eastern Mediterranean Route by land (EMR by land). We have seen that in 2014, the CMR was slightly the favorite one. However, during the European refugee crisis of 2015, Syrians avoided the CMR and the EMR by sea became by far the most popular, overshadowing the refugee volumes of the other routes.

Table 4 illustrates this with the results from this case study. In this table we present the real options values for the three main routes (CMR, EMR by sea, EMR by land) for 2014 and 2015 (see columns 3 resp. 5 in Table 4). These values were calculated above in Tables 1 to 3. For instance, the real option value of the EMR by sea route goes up from 3,179 euro in 2014 to 3,809 in 2015. The real option model indicates that the most valuable route in 2014 is the CMR route, while in 2015 it is the EMR by sea (indicated in bold in Table 4). We also collected data on the actual amount of Syrian refugees per route as detected by Frontex, the European Border and Coast Guard Agency in the years 2014 and 2015. The most popular route in 2014 was the CMR route which accounted for 50% of the Syrian refugees, while this changed to the EMR by sea route in 2015 (82% of Syrian refugees) (See columns 3 resp. 5 in Table 4). One can observe that the relative attractiveness between the routes is reflected in the real options values. We

**Table 4. Real options values of three main routes and relative amount of illegal border crossings.**

|  | 2014 | | 2015 | |
|---|---|---|---|---|
|  | **Real options value** | **Relative illegal border crossings** | **Real options value** | **Relative illegal border crossings** |
| CMR | **10,388** | **50%** | -113 | 1% |
| EMR by sea | 3,179 | 34% | **3,809** | **82%** |
| EMR by land | 674 | 6% | 577 | 1% |

CMR–Central Mediterranean Route; EMR by sea–Eastern Mediterranean Route by sea; EMR by land–Eastern Mediterranean Route by land. Relative illegal border crossings of Syrians only as detected by Frontex in 2014 or 2015. Real option value in euros as calculated in Tables 1–3; highest option value and highest amount of illegal border crossings presented in bold.

Source: prepared by the authors.

could for example see that the CMR has the highest real option value in 2014, which is consistent with the fact that this was the most popular route. In 2015 the highest option value shifts towards the EMR by sea route, which is again the most popular route on the field in that year, accounting for 82% of the illegal crossings.

As the real option value captures the attractiveness of a certain route, policy makers can also focus on one of more value-drivers of the real option model to actively manage refugee flows. Our model shows that the attractiveness of a route depends on the value-drivers of the real option model, such as the benefits of travel, the cost of travel, the length of travel, and the probability of travel, among others (see, for instance, the value-drivers in Table 1). By altering the value of one of the value drivers, policy makers have the ability to influence the real option value of a particular route. Put differently, they have the ability to influence the attractiveness of this route and they are able to steer the route choice of the refugee. This active management can occur both at the supranational EU level as well at the individual country level.

At the supranational level, the EU has the ability to make one route more or less attractive than another in order to better allocate the migrants throughout Europe and reduce the pressure on the frontline states. To change the real options value, the EU policy level has to focus on certain value-drivers of the real option model that impact the options value, and hence change the attractiveness of a certain route. For instance, if the strategy is aimed at reducing the attractiveness, a potential value-driver is to increase the travel costs, while the opposite is true if one would like to increase the attractiveness of a certain route. To achieve the latter, Frontex could decrease the costs of travel by organizing free transport to the assigned country in the context of the EU Relocation plan. Relocation refers to the transfer of refugees who are in clear need of international protection from one EU state to another EU state. By lowering the travel costs for refugees, a particular route will become more attractive, while another route will become less attractive. In this way, Frontex can actively try to manage to divert refugee flows from member states under too much pressure to member states with more capacity to handle a certain amount of refugees.

At the individual country level, many countries do not have the capacity and resources to host a large amount of refugees. Therefore, these countries try to decrease the attractiveness of the routes to and through their territories. For instance, individual countries have the possibility to decrease the real option value by lowering the probability of arrival. A country can achieve this by introducing border checks or by building border fences.

As policy actions to impact one or more value-drivers of the real options model can occur both at the EU level and at the level of individual countries, such actions can reinforce or counteract with each other. It is important to note here that this is not a shortcoming of the model, but merely the consequence of political choices with the different EU member states. Hence, on the contrary, our real options model is well-suited to provide further insights in the anticipated (joint) effects of different policy actions.

## 3.4. Limitations

While our case study shows preliminary evidence for the fitness of a compound real options framework to model refugee routes, this section points towards some limitations. A first limitation is related to the case study approach as we focus on the average Syrian male refugee as the focal decision-maker [44]. It is unclear whether our conclusion can be transposed to other refugees which are non-Syrian or non-male. Although we conjecture that the model could also be applied to different situations and therefore allowing for heterogeneity among the refugee population, future research needs to confirm our model in different contexts. For instance, family members who travel together and also have a mutual interdependence: if one of the

family members does not want or is not able to travel, this could also affect the choices of the other family members. Another example could be an individual refugee who chooses to stay in a transit country instead of traveling to the desired final destination. Although we did not consider these situations, the compound option model could easily incorporate such features. Furthermore, the country of origin could affect the decision of the refugee regarding choice of the final destination. During the refugee crisis, Germany declared an open-door policy for Syrian refugees, while its policy might not be so generous to refugees from countries that are not at war. We invite researchers to replicate our study for refugees from other countries or continents. Analyzing different type of refugees could lead to different input values for the model, such as duration of the journey, probability of arrival, costs and benefits and uncertainty in benefits, which in turn would deliver a different option value in comparison to that of the case of the Syrian male refugee.

Second, it is challenging to determine the exact value for certain input parameters, because data is not available, unreliable or not quantifiable. For example, in case it is unknown how many refugees travel from one to another country, it would be hard to determine the probability of arrival. Furthermore, there is also little information available about people who have made the trip multiple times, which prevents to take into account issues with double counting. More reliable microdata would lead to better results, therefore it would be very helpful to have access to better finer-grained data sources.

A third potential limitation is the idea that it may not be realistic to expect a refugee, whose major concern is survival, to perform a rational cost and benefit analysis in case of forced migration. However, even though socio-economic actors do not always conduct an explicit cost-benefit analysis, the framework accurately predicts human behavior. A good example is criminal behavior. Even though not all criminal perform an explicit analysis, economic models accurately predict criminal behavior [56]. The forced character of refugees also poses potential problems with regard to compound option analogy in case of "timing". According to real options logic, the more time there is to make a decision, the more the option is worth. In case of involuntary migration, such as during the first part of the journey of the Syrian refugee, option logic is applicable to a lesser extent.

A fourth and final limitation is the feasibility of using the real options analysis in monitoring and steering policy measures. While the impact of policy makers on the attractiveness of certain routes by leveraging one or more value-drivers of the real option model seems straightforward in theory, one can imagine numerous challenges in practice. Many times precise data might be missing, making it more difficult to calculate the precise real option value. However, we conjecture that in many case applying the real option logic to route choice of refugees is often more important than the exact valuation of such routes. To gain better insight in the pros and cons of implementing our model in practice, a study using our model to retrospectively assess the policies implemented in Europe following the 2015 Syrian refugee flows could be very useful. This will be left for future research. At this stage, we invite Frontex to start collecting data on the different value-drivers of the real option model to assist more efficient future decision-making.

## 4. Conclusion

During the refugee crisis the unprecedented number of refugee arrivals in Europe has created new and complex challenges for the EU, member states, and the international community at large. This calls for a tool to analyze refugee flows. In this article we have used compound real options analysis to model refugee flows. A refugee has to cope with many uncertainties, such as for example the changing unilateral and EU-wide legislation, and in more concrete terms,

the uncertainty regarding the possibility to obtain a seat on a boat. At the same time, the refugee still has the possibility to make an informed decision regarding factors such as the desired final destination, the expected costs and benefits, the travel time and the uncertainty of reaching their final destination. Through modern channels, such as Facebook, a refugee still could have the possibility to make a reasonably informed decision and to opt for one route over another.

In order to be able to model refugee flows we have used a compound real option model. We have illustrated on the basis of a case-study how refugee flows could be modeled through the use of a compound option models by quantifying the attractiveness of a refugee route relative to other routes. In this case-study we have demonstrated that the popularity of different routes, and thereby the dynamics of the decision-making process of the average refugee, runs parallel with their real options values. The effects of the changing conditions in the countries of origin, transit and destination could be calculated through a real options framework. For example, we have noticed that the closure of Libya, translated to an enormous decrease of the probability of arrival from Syria to Libya, indeed had a strong negative impact on the real options value. This in turn corresponds to the enormous decrease in refugee volumes to Libya. Policy makers could use this framework prior to the implementation of policy measures. Real options analysis offers the possibility to provide an ex-ante estimation of the effects on refugee volumes of policy measures. Moreover, (near) real-time refugee data would also provide the possibility to monitor and steer the influence of policy measures accordingly.

In general, there are obviously more players involved than just the policy maker. Measures by an EU policy maker call for reactions by the other actors, such as the refugees, individual countries and the smuggler, who plays an important role as a facilitator of the migration. In order to present a complete picture of effective refugee policy, it is also important to take into account this interaction into the analysis. Otherwise, there could be the risk that a policy measure works counter-productive and outdrives the intended objectives, such as that persons will disappear into illegality, but still do come to Europe. After all, refugee flows are said to be unstoppable [12], with (more stringent) policy measures merely diverting flows, possibly causing additional unintended negative effects.

## Author Contributions

**Conceptualization:** Peter-Jan Engelen, Danny Cassimon.

**Data curation:** Linda Peters, Peter-Jan Engelen.

**Formal analysis:** Peter-Jan Engelen, Danny Cassimon.

**Investigation:** Linda Peters.

**Methodology:** Linda Peters, Peter-Jan Engelen, Danny Cassimon.

**Software:** Peter-Jan Engelen.

**Validation:** Danny Cassimon.

**Writing – original draft:** Linda Peters, Peter-Jan Engelen.

**Writing – review & editing:** Peter-Jan Engelen, Danny Cassimon.

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
