## [Decision Letter · Decision Letter 0]

8 Mar 2023

PONE-D-22-30914Explaining Refugee Flows. Understanding the 2015 European refugee crisis through a real option lensPLOS ONE

Dear Dr. Cassimon,

Thank you for submitting your manuscript to PLOS ONE. After careful consideration, we feel that it has merit but does not fully meet PLOS ONE’s publication criteria as it currently stands. Therefore, we invite you to submit a revised version of the manuscript that addresses the points raised during the review process.

We look forward to receiving your revised manuscript.

Kind regards,

Adetayo Olorunlana, Ph.D.

Academic Editor

PLOS ONE

2. We note that [Figure 2] in your submission contain [map/satellite] images which may be copyrighted. All PLOS content is published under the Creative Commons Attribution License (CC BY 4.0), which means that the manuscript, images, and Supporting Information files will be freely available online, and any third party is permitted to access, download, copy, distribute, and use these materials in any way, even commercially, with proper attribution. For these reasons, we cannot publish previously copyrighted maps or satellite images created

using proprietary data, such as Google software (Google Maps, Street View, and Earth). For more information, see our copyright guidelines: http://journals.plos.org/plosone/s/licenses-and-copyright.

a. You may seek permission from the original copyright holder of Figure(s) [#] to publish the content specifically under the CC BY 4.0 license. 

Natural Earth (public domain): http://www.naturalearthdata.com/.

Reviewers' comments:

Reviewer's Responses to Questions

**Comments to the Author**

1. Is the manuscript technically sound, and do the data support the conclusions?

Reviewer #1: Yes

Reviewer #2: Partly

Reviewer #3: Yes

2. Has the statistical analysis been performed appropriately and rigorously? 

Reviewer #1: Yes

Reviewer #2: N/A

Reviewer #3: Yes

3. Have the authors made all data underlying the findings in their manuscript fully available?

Reviewer #1: Yes

Reviewer #2: Yes

Reviewer #3: Yes

4. Is the manuscript presented in an intelligible fashion and written in standard English?

Reviewer #1: Yes

Reviewer #2: Yes

Reviewer #3: Yes

5. Review Comments to the Author

Reviewer #1: I like the paper a lot. The RO approach is the right choice for assessing travel routes as nicely pointed out by the authors. I have a few minor comments. On page 6 when you refer to rational choice, the wording is not precise enough. Observed choices are rational. Refugees are making rational choices based on the information available to them. The wording needs to be adjusted. Top of page 7, not only uncertainty is necessary but also irreversibility for creating a real option value. On page 8, the explanation for V* should be adjusted. This is the threshold level for immediate start or continuation of the travel. many readers might not be familiar with language used in finance. This can be added in brackets. For the case studies I miss reference to the irreversible costs (exercise price) applied in the different cases. A change in those prices should have an effect on the RO value and be discussed.

Reviewer #2: This manuscript tries to use a real options framework to model the 2015 refugee flows into Europe. Although I see very little value of this in the current context of migration in Europe in 2023, the methodology used is interesting.

However, the authors should stress very strongly the fact that it is very questionable whether this model applies to other types of refugees (non-Syrians, non-males).

Moreover, the context this model can be used in policy making is also unclear.

Is it at the European level or at the EU country level? Both possibilities present numerous challenges. It would also be useful to clarify what is meant by policy making and how closely could the real options analysis could ever predict the effects of policy measures on refugees. This population group has a well established mobility, different behavioral patterns, traits and the quantitative data officially available to capture their way in Europe are notoriously incomplete.

I suggest to introduce a paragraph in the limitations section outlining a more realistic approach on the feasibility of using the real options analysis in monitoring and steering policy measures. Perhaps to suggest using this analysis to retrospectively assess the policies implemented in Europe following the 2015 Syrian refugee flows.

Reviewer #3: This manuscript is important and presents and discusses a topical subject of the time. It is well conceptualized, written and analyzed and presented in a logical and flowing manner. The statistical and economic assumptions and analyses are satisfactorily well done. However there are a couple of edits and corrections that are required.

1) there are numerous typographical, grammatical errors that need to be corrected. In several paragraphs the tenses used are in present tense instead of past tense for events that have occurred in the past. These are in several parts of the manuscript and should be wholesomely corrected throughout the entire manuscript.

2)In a beginning paragraphs, .....certain routes are more interesting... the word interesting could be replaced with more preferred..

3) Another paragraph..................the sentence starting. A journey of a such a refugee... delete the initial a

4) The sentence above Fig1 The optimal nature........doesn't real well and is not clear, i.e. ....that is options on options, and its value.... this is not clear.

5) Under table 1> Value drivers ...The sentence below the journey from Syria to Libya in 2014 is mostly travelled by road...grammatical errors

6) Under section 3.3 Discussion, while the fist paragraph is good discussion of the study findings, the second paragraph is a very unsatisfactory explanation of the findings and the routs in a very unattractive way, I suggest that it gets completely overhauled and re written better.

7) Conclusion, the last last paragraph. In general, there are.... ....players involved THEN just .. please replace the then by THAN just...

6. PLOS authors have the option to publish the peer review history of their article (what does this mean?). If published, this will include your full peer review and any attached files.

Reviewer #1: No

Reviewer #2: No

Reviewer #3: **Yes: **Christopher Garimoi Orach

---

## [Author Response · Author response to Decision Letter 0]

27 Mar 2023

Many thanks for the invitation to revise and resubmit our paper entitled “Explaining Refugee Flows. Understanding the 2015 European refugee crisis through a real option lens” (PONE-D-22-30914) to PLoS One. 

We studied the reviewers’ feedback carefully and have worked hard to address all of the issues and concerns they raised. Overall, we have adjusted the wording (R1C1-4, R3C2-5), corrected typos (R3C1, R3C3, R3C7), incorporated better discussions (R3C6), and inserted new paragraphs on policy implications (R2C2) and limitations (R2C1, R2C2).

We also updated Figure 2 (map) with non-copyrighted material using your suggestion of USGS National Map Viewer (public domain).

We now provide a detailed point-by-point response to the queries and concerns raised by the reviewers. We would be grateful for a further opportunity to improve the manuscript if our responses do not meet expectations. 

REVIEWER #1: 

I like the paper a lot. The RO approach is the right choice for assessing travel routes as nicely pointed out by the authors. I have a few minor comments. 

Answer: Thank you very much for appreciating our manuscript.

Comment 1: 

On page 6 when you refer to rational choice, the wording is not precise enough. Observed choices are rational. Refugees are making rational choices based on the information available to them. The wording needs to be adjusted. 

Answer: We adjusted the wording.

Comment 2: 

Top of page 7, not only uncertainty is necessary but also irreversibility for creating a real option value. 

Answer: We added this in the text to make this point more clear.

Comment 3: 

On page 8, the explanation for V* should be adjusted. This is the threshold level for immediate start or continuation of the travel. Many readers might not be familiar with language used in finance. This can be added in brackets. 

Answer: Thank you for this suggestion. We added this explanation between brackets for readers less familiar with finance.

Comment 4: 

For the case studies I miss reference to the irreversible costs (exercise price) applied in the different cases. A change in those prices should have an effect on the RO value and be discussed.

Answer: We included this reference in the text. We also made extra calculations showing the impact of the changes in the irreversible costs in isolation and included them in the text (see page 18).

REVIEWER #2: 

This manuscript tries to use a real options framework to model the 2015 refugee flows into Europe. Although I see very little value of this in the current context of migration in Europe in 2023, the methodology used is interesting.

Answer: Thank you for appreciating our methodology in the context of refugee flows.

Comment 1:

However, the authors should stress very strongly the fact that it is very questionable whether this model applies to other types of refugees (non-Syrians, non-males).

Answer: Thank you for pointing out this potential limitation of our case study. We incorporated your comments as the primary limitation in Section 3.4. We stress that our case study conclusions cannot be transposed automatically to other types of refugees and we invite future research to replicate our model for non-Syrians and non-male refugees.

Comment 2:

Moreover, the context this model can be used in policy making is also unclear. 

Is it at the European level or at the EU country level? Both possibilities present numerous challenges. It would also be useful to clarify what is meant by policy making and how closely could the real options analysis could ever predict the effects of policy measures on refugees. This population group has a well-established mobility, different behavioral patterns, traits and the quantitative data officially available to capture their way in Europe are notoriously incomplete.

Answer: Thank you very much for raising this point. We included several new paragraphs in the Discussion (Section 3.3) explaining how policy makers can impact the real option value and, hence, the attractiveness of a certain route. We distinguish between the EU level and the individual country level. We provide two examples of such actions. A complete analysis of all potential actions within the EU refugee crisis is outside the confines of this article and will be left for future research. In fact, your valuable comment inspired us to take up this issue in a follow-up paper.

Comment 3:

I suggest to introduce a paragraph in the limitations section outlining a more realistic approach on the feasibility of using the real options analysis in monitoring and steering policy measures. Perhaps to suggest using this analysis to retrospectively assess the policies implemented in Europe following the 2015 Syrian refugee flows.

Answer: We included a new paragraph in the Limitations (Section 3.4) discussing potential limits in using real option analysis in policy measures. Thank you for your suggestion of the retrospective study, an idea we would like to take up in a follow-up paper.

REVIEWER #3: 

This manuscript is important and presents and discusses a topical subject of the time. It is well conceptualized, written and analyzed and presented in a logical and flowing manner. The statistical and economic assumptions and analyses are satisfactorily well done. However there are a couple of edits and corrections that are required.

Answer: Thank you for appreciating our research efforts.

Comment 1:

There are numerous typographical, grammatical errors that need to be corrected. In several paragraphs the tenses used are in present tense instead of past tense for events that have occurred in the past. These are in several parts of the manuscript and should be wholesomely corrected throughout the entire manuscript.

Answer: Many thanks for reading our manuscript so carefully. We edited the text.

Comment 2:

In a beginning paragraphs, .....certain routes are more interesting... the word interesting could be replaced with more preferred..

Answer: Thank you for your suggestion. We changed the wording.

Comment 3:

Another paragraph..................the sentence starting. A journey of a such a refugee... delete the initial a

Answer: Apologies for the typo, we adjusted this.

Comment 4:

The sentence above Fig1 The optimal nature........doesn't real well and is not clear, i.e. ....that is options on options, and its value.... this is not clear.

Answer: We rephrased the sentence to improve the readability.

Comment 5:

Under table 1> Value drivers ...The sentence below the journey from Syria to Libya in 2014 is mostly travelled by road...grammatical errors

Answer: We adjusted the wording to make the sentence more clear.

Comment 6:

Under section 3.3 Discussion, while the first paragraph is good discussion of the study findings, the second paragraph is a very unsatisfactory explanation of the findings and the routs in a very unattractive way, I suggest that it gets completely overhauled and re written better.

Answer: Following your suggestion we have rewritten the second paragraph completely. We merged Tables 4 and 5 into one new Table 4 and indicated the highest option values and most attractive routes in bold to more clearly show the parallel between both. We also rewrote the accompanying text to present our findings more clearly.

Comment 7:

Conclusion, the last last paragraph. In general, there are.... ....players involved THEN just .. please replace the then by THAN just...

Answer: We adjusted the typo.

---

## [Editor Report · Decision Letter 1]

29 Mar 2023

Explaining Refugee Flows. Understanding the 2015 European refugee crisis through a real options lens

PONE-D-22-30914R1

Dear Dr. Cassimon,

We’re pleased to inform you that your manuscript has been judged scientifically suitable for publication and will be formally accepted for publication once it meets all outstanding technical requirements.

Kind regards,

Adetayo Olorunlana, Ph.D.

Academic Editor

PLOS ONE
---

## [Editor Report · Acceptance letter]

11 Apr 2023

PONE-D-22-30914R1 

Explaining Refugee Flows.
 
Understanding the 2015 European refugee crisis through a real options lens 

Dear Dr. Cassimon:

I'm pleased to inform you that your manuscript has been deemed suitable for publication in PLOS ONE. Congratulations! Your manuscript is now with our production department. 

Kind regards, 

on behalf of

Associate Professor Adetayo Olorunlana 

Academic Editor

PLOS ONE